# Score-based Continuous-time Discrete Diffusion Models

**Haoran Sun**[*]
Georgia Tech
hsun349@gatech.edu

**Lijun Yu**
Carnegie Mellon University
lijun@cmu.edu

**Bo Dai**
Google Research; Georgia Tech
bodai@google.com

**Dale Schuurmans**
Google Research; University of Alberta
schuurmans@google.com

**Hanjun Dai**
Google Research
hadai@google.com

## Abstract

Score-based modeling through stochastic differential equations (SDEs) has provided a new perspective on diffusion models, and demonstrated superior performance on continuous data. However, the gradient of the log-likelihood function, *i.e.*, the score function, is not properly defined for discrete spaces. This makes it non-trivial to adapt the score-based modeling to categorical data. In this paper, we extend diffusion models to discrete variables by introducing a stochastic jump process where the reverse process denoises via a continuous-time Markov chain. This formulation admits an analytical simulation during backward sampling. To learn the reverse process, we extend score matching to general categorical data, and show that an unbiased estimator can be obtained via simple matching of the conditional marginal distributions. We demonstrate the effectiveness of the proposed method on a set of synthetic and real-world music and image benchmarks.

## 1 Introduction

Diffusion models (Sohl-Dickstein et al., 2015; Ho et al., 2020) have emerged as an important technique for data distribution modeling, where a data-corrupting forward process is coupled with a denoising reverse process to simulate a diffusion relationship between the data distribution and an uninformed source. Such models admit stable learning procedures and have demonstrated superior performance on continuous data modeling in challenging scenarios (Dhariwal & Nichol, 2021), leading to rapidly increasing popularity. Song et al. (2020) established a stochastic differential equation view of diffusion models by forming the limit of finer corruption and denoising steps in the forward and backward processes, rendering a continuum of distributions. This perspective has provided a unified framework under a new score-based learning objective, and inspired a variety of simulation methods for efficient sampling and inference (De Bortoli et al., 2021; Zhang & Chen, 2022).

Given the advantages of diffusion models in terms of flexibility, learning tractability, and sampling, there have been several attempts to extend the approach to discrete data. Recent attempts have investigated alternative corruption operations for discrete data in the forward process, yielding promising results (Hoogeboom et al., 2021b;a; Austin et al., 2021). However, these extensions still execute a finite sequence of corruption and restoration steps, and remain restricted to a fixed reverse sampling strategy that can be sub-optimal. To overcome this limitation, we investigate whether a *continuous-time* discrete diffusion formulation might admit more effective estimation and generation.

Such an extension is highly non-trivial however. The continuous-time diffusion framework is based on a stochastic differential equation (SDE) with respect to the score function, which itself is the gradient of the log-likelihood with respect to a continuous variable. Although this can be used to characterize a continuum of infinitesimally evolving distributions over a continuous space, such a formulation no longer exists for discrete variables, since the gradient of the log-likelihood does not exist with respect to a discrete variable. Recently, Campbell et al. (2022) made significant progress in

---

[*]Work done during an internship at Google.

closing this gap by proposing a discrete data distribution that evolves in a continuous-time Markov chain. With this formulation they were able to approximate maximum likelihood training with an evidence lower bound (ELBO) surrogate and using a predictor-corrector sampling scheme to generalize some of the benefits of continuous-time modeling to a discrete space.

However, a limitation of this previous work is the reliance on an ELBO approximation to the MLE when it is known that score-based learning yields superior estimation quality (when it can be applied) due to its unbiasedness (Song et al., 2020). Of course, developing an analog of the score matching for discrete spaces is non-trivial due to non-differentiability. Nevertheless, a score function for a discrete random variable cannot be arbitrary: whenever two score functions match over a space, their corresponding distributions should also match; the score function should characterize the direction of infinitesimal evolution of a discrete distribution; and finally the score should enable tractable score-based estimation. In this paper, we investigate the design of such score functions for discrete spaces achieving these desired properties, and provide corresponding learning and sampling methods. In addition, we complete the agenda for continuous time diffusion in discrete spaces by formulating a coherent SDE in terms of stochastic jump processes. The main contributions are:

- We extend the definition of score functions to generic categorical discrete variables, and derive a continuous-time discrete diffusion model via continuous time Markov chain in Section 3.1;
- We derive a score-based objective called categorical ratio matching for estimating the proposed model in Section 3.2, which can be tractably optimized, showing that a previous proposal for binary discrete data (Hyvärinen, 2007) can be obtained as a special case;
- In Section 4, we develop a numerical simulation technique for reverse sampling, then provide an analytical sampling method based on implicit modeling of the conditional marginals;
- We discuss three architectural choices and present a novel "hollow" neural model in Section 5;
- We evaluate the proposed SDDM on a set of synthetic and real-world music and image benchmarks, achieving promising results in Section 6.

## 2 BACKGROUND

Diffusion models (Sohl-Dickstein et al., 2015) are characterized by a forward Markov process that transforms an observation $x_0 \sim \pi_{data}(x_0)$ to a reference distribution $x_T \sim q_T(x_T)$, and a backward process, which is also Markovian, that recovers the data distribution from $x_T$. Specifically, the forward process is defined through simple corruption operations, $q_{t+1|t}$. For continuous data the corruption kernel usually adds Gaussian noise (Ho et al., 2020); whereas for discrete data, where $x_0 \in \mathcal{X}$ with finite cardinality $|\mathcal{X}|$, the corruption kernel can be uniform, discrete Gaussian, or some other choice (Austin et al., 2021; Johnson et al., 2021). Given the corruption kernel, after $T$-steps, the forward process forms a joint distribution,

$$q_{0:T}(x_{0:T}) = \pi_{data}(x_0) \prod_{t=0}^{T-1} q_{t+1|t}(x_{t+1}|x_t). \tag{1}$$

The backward process can be derived from the joint distribution via Bayes' rule,

$$q_{0:T}(x_{0:T}) = q_T(x_T) \prod_{t=0}^{T-1} q_{t|t+1}(x_t|x_{t+1}), \quad q_{t|t+1}(x_t|x_{t+1}) = \frac{q_{t+1|t}(x_{t+1}|x_t)q_t(x_t)}{q_{t+1}(x_{t+1})}, \tag{2}$$

where $q_t(x_t)$ denotes the marginal distribution and the prior $q_T$ is usually a simple distribution. Typically the backward kernel $q_{t|t+1}(x_t|x_{k+1})$ is intractable; thus it is usually parameterized by a neural network, denoted $p_{t|t+1}^\theta$, and learned via ELBO (Sohl-Dickstein et al., 2015; Ho et al., 2020; Austin et al., 2021) or score-matching (Song & Ermon, 2019). Due to the structure of the joint distribution (1) and (2), the ELBO admits a particularly simple formulation as

$$\ell_{vb} = \mathbb{E}_{\pi_{data}} \left[ D_{KL}(q_{T|0}||q_T) + \sum_{t=1}^{T-1} \mathbb{E}_{q_{t|0}} \left[ D_{KL}\left(q_{t|t+1,0}||p_{t|t+1}^\theta\right) \right] - \mathbb{E}_{q_{1|0}} \left[ \log p_{0|1}^\theta(x_0|x_1) \right] \right], \tag{3}$$

which is applicable for both continuous and discrete diffusion model learning. For continuous variables with Gaussian corruptions $q(x_{t+1}|x_t) = \mathcal{N}\left(x_{t+1}; \sqrt{1-\beta_{t+1}}x_t, \beta_{t+1}I\right)$, and a backward kernel $p^\theta(x_t|x_{t+1}) = \mathcal{N}\left(x_t; \frac{1}{\sqrt{1-\beta_{t+1}}}\left(x_{t+1} + \beta_{t+1}r_t^\theta(x_{t+1})\right), \beta_{t+1}I\right)$, such that $r_t^\theta$ is learned as a neural network and $\beta_t$ is a predefined variance schedule, the ELBO (3) can be rewritten as

$$\ell_{vb} = \sum_{t=0}^{T-1} (1-\alpha_t) \mathbb{E}_{\pi_{data}} \mathbb{E}_{p_{\alpha_t}(x'|x)} \left[ \left\| r_t^\theta(x') - \nabla_{x'} \log p_{\alpha_t}(x'|x) \right\|_2^2 \right], \tag{4}$$

where $\alpha_t = \prod_{t=0}^{t-1}(1 - \beta_t)$. The Equation 4 is highly related to score-matching (Hyvärinen & Dayan, 2005) with a score function defined as $\nabla_x \log p_{t+1|t}(x_{t+1}|x_t)$.

Reverse sampling in discrete-time models is somewhat restricted by the forward process. Therefore, continuous-time diffusion models have been constructed with $t$ indexed from $[0, T]$. Song et al. (2020) develop an SDE perspective and define a diffusion process with respect to the score as

$$\mathrm{d}x = f(x, t)\, \mathrm{d}t + g(t)\mathrm{d}\boldsymbol{w}, \qquad\qquad \text{forward SDE,} \qquad (5)$$

$$\mathrm{d}x = \left[f(x, t) - g^2(t)\nabla_x \log p_t(x)\mathrm{d}t\right] + g(t)\mathrm{d}\bar{\boldsymbol{w}}, \qquad \text{reverse SDE,} \qquad (6)$$

where $\boldsymbol{w}$ and $\bar{\boldsymbol{w}}$ are standard Wiener processes, $f(x, t)$ is a vector-valued drift function, and $g(t)$ is a scalar-valued diffusion coefficient. A score-based learning method can be easily derived as an extension of (4). However, the score function $\nabla_x \log p_t(x)$ is not defined for a discrete variable, and the SDE above no longer suffices to characterize a continuous-time extension for discrete diffusion.

## 3 CONTINUOUS TIME DISCRETE SCORE MATCHING

Although Campbell et al. (2022) bypass the score function in a continuous-time extension, by leveraging a stochastic process view of the ELBO approximation, we will instead focus on a score-based extension for continuous-time discrete diffusion.

### 3.1 CONTINUOUS TIME MODELING

Consider the finite discrete state space $\mathcal{X} = \mathcal{C}^D$, where $\mathcal{C} = \{1, 2, ..., C\}$ is a code book. To generalize score matching from a continuous space $\mathbb{R}^n$ to discrete space $\mathcal{X}$, we first model the forward process as a continuous time Markov chain $\{X_t\}_{t \in [0,T]}$, whose transition probability is characterized by rate matrices $Q_t \in \mathbb{R}^{|\mathcal{X}| \times |\mathcal{X}|}$. In particular, if we let $q$ denote the distribution for the forward process $X_t$, the transition probability will satisfy the Kolmogorov forward equation:

$$\tfrac{d}{dt}q_{t|s}(x_t|x_s) = \sum_{x \in \mathcal{X}} q_{t|s}(x|x_s)Q_t(x, x_t), \quad s < t \qquad (7)$$

If the forward process starts at the target distribution $q_0 = \pi_{\text{data}}$, the marginal at time $t$ has the form:

$$q_t(x_t) = \int_{\mathcal{X}} \pi_{\text{data}}(x_0)q_{t|0}(x_t|x_0)dx_0 \qquad (8)$$

By properly choosing the rate matrices $Q_t$, we can achieve a final distribution close to a known tractable distribution $q_T \approx \pi_{\text{ref}}$. Then, the reverse time process $\overline{X}_t = X_{T-t}$ can be expressed as a generative process from the reference distribution $\pi_{\text{ref}}$ to the target distribution $\pi_{\text{data}}$ (Anderson & Rhodes, 1983; Van Handel, 2007).

**Proposition 3.1.** *The reverse time process $\overline{X}_t$ of the continuous time Markov chain $X_t$ is also a Markov process, whose transition probabilities $q_{s|t}(\cdot|\cdot)$ for $s < t$ satisfy:*

$$q_{s|t}(x_s|x_t) = \tfrac{q_s(x_s)}{q_t(x_t)}q_{t|s}(x_t|x_s), \quad s < t \qquad (9)$$

Since we are considering a finite space $\mathcal{X}$, the reverse time process $\overline{X}$ is uniquely determined by its rate matrices, which we denote as $R_t$. Using the transition in Equation 9, we have the following.

**Proposition 3.2.** *For a continuous time Markov chain $\{X_t\}_{t \in [0,T]}$ with distribution $q$ and rate matrices $Q_t$, the rate matrices $R_t$ for the reverse process satisfy:*

$$R_t(x, y) = \frac{q_t(y)}{q_t(x)}Q_t(y, x) \qquad (10)$$

Equation 10 provides a closed form expression for the reverse time rate $R_t$. Therefore, once we know the ratio $q_t(y)/q_t(x)$, we can obtain the generative flow towards $\pi_{\text{data}}$.

### 3.2 CATEGORICAL RATIO MATCHING

In general, the ratio $q_t(y)/q_t(x)$ in Equation 10 is intractable. This ratio behaves analogously to the score function $\nabla \log \pi(x)$ in Equation 6. Such a connection inspires learning the ratio $q_t(y)/q_t(x)$ in

a similar manner to learning the score function for diffusion models in continuous spaces. For binary discrete variable, Hyvärinen (2007) proposed to match the ratio $\pi(X^{\backslash d}, X^d = c)/\pi(X^{\backslash d}, X^d = 1 - c)$ as an extension for score matching. In this work, we consider the more general categorical case, where neither the score function $\nabla \log \pi(x)$ nor the binary ratio is well defined. The generalization relies on the singleton conditional distribution:

$$\pi(X^d = c | x^{\backslash d}) = \pi(x^{\backslash d}, X^d = c) / \sum_{c' \in \mathcal{C}} \pi(x^{\backslash d}, X^d = c') \tag{11}$$

yielding a score function in categorical space we seek to match. The sufficiency of matching Equation 11 is guaranteed by the property that the joint distribution is completely determined by its singleton conditional distributions (Brook, 1964; Lyu, 2012).

**Proposition 3.3.** *Consider random variables $X = (X_1, ..., X_D) \in \mathcal{X}$, and two probability distributions $\pi_1$, $\pi_2$. We have $\pi_1 = \pi_2$, if and only if their conditional distributions are equal $\pi_1(X^d = x^d | x^{\backslash d}) = \pi_2(X^d = x^d | x^{\backslash d})$, for any $x \in \mathcal{X}$ and $d = 1, ..., D$.*

Going back to the ratio $q_t(y)/q_t(x)$ in reverse time rate Equation 10, Proposition 3.3 tells us we can recover the probability ratio by employing a time dependent neural network $p_t(\cdot; \theta)$ to match the conditional distribution,

$$p_t(X^d | x^{\backslash d}; \theta) \approx q_t(X^d | x^{\backslash d}) \Rightarrow \frac{q_t(y^d, x^{\backslash d})}{q_t(x^d, x^{\backslash d})} = \frac{q_t(X_t^d = y^d | x^{\backslash d})}{q_t(X_t^d = x^d | x^{\backslash d})} \approx \frac{p_t(X_t^d = y^d | x^{\backslash d}; \theta)}{p_t(X_t^d = x^d | x^{\backslash d}; \theta)} \tag{12}$$

To train $p_t(\cdot; \theta)$, we minimize the expected cross entropy with respect to the conditional distributions along the forward process:

$$\theta^* = \arg \min_\theta \int_0^T \sum_{x_t \in \mathcal{X}} q_t(x_t) \left[ \sum_{d=1}^D \left( - \sum_{c \in \mathcal{C}} q_t(X_t^d = c | x_t^{\backslash d}) \log p_t(X_t^d = c | x_t^{\backslash d}; \theta) \right) \right] dt \tag{13}$$

where the loss is minimized if $p_t(X_t^d | x_t^{\backslash d}; \theta) \equiv q_t(X_t^d | x_t^{\backslash d})$. This loss function matches the ratio in Equation 11, hence we name it as categorical ratio matching as a generalization of Hyvärinen (2007). Since the spirit of this loss function is the same as the score matching in continuous space, we also interchangeably call this method as categorical score matching. In Equation 13, the expectation over $q_t(\cdot)$ can be efficiently estimated via Monte Carlo sampling. However, the conditional distribution $q_t(X_t^d = c | x_t^{\backslash d})$ is intractable, which brings difficulty to the training. Fortunately, using the property of conditional distribution, we can avoid computing this intractable term in the loss function.

**Proposition 3.4.** *The categorical ratio matching loss function in Equation 13 can be simplified as:*

$$\theta^* = \arg \min_\theta \int_0^T \sum_{x_t \in \mathcal{X}} q_t(x_t) \left[ \sum_{d=1}^D - \log p_t(X^d = x_t^d | x_t^{\backslash d}; \theta) \right] dt \tag{14}$$

Using this surprisingly neat loss function in Equation 14, we can efficiently learn the conditional distribution. The learned $p_t(X_t^d | x_t^{\backslash d}; \theta)$ determines a reverse process, and we use $p(\cdot; \theta)$ to denote its joint distribution in order to distinguish with the true reverse process. We will sometimes drop the $\theta$ if it does not create ambiguity.

## 4 CONTINUOUS TIME DISCRETE SAMPLING

We next develop and efficient sampling scheme for the proposed diffusion model.

### 4.1 CONTINUOUS TIME SIMULATION FOR FORWARD PROCESS

In a continuous time Markov chain, the transition matrix $q_{t|s}(\cdot | \cdot)$ from time $s$ to time $t$ in the forward process can be obtained by solving the ODE in Equation 7. For general rate matrices $Q_t \in \mathbb{R}^{|\mathcal{X}| \times |\mathcal{X}|}$, solving the ODE is intractable. Therefore, we follow standard practice in diffusion models (Austin et al., 2021; Campbell et al., 2022) and approximate the forward process by factorizing $X_t = (X_t^1, ..., X_t^D)$ where each sub-process $X_t^d$ propagates independently. Furthermore, we also define the sub-rate matrix $Q_t^d = Q\beta(t)$ in terms of a fixed base rate $Q = P\Lambda P^{-1} \in \mathbb{R}^{C \times C}$

and time schedule function $\beta(t)$; see design of $\beta(t)$ in Appendix C.1. In this way, the sub-transition matrix in each dimension can then be easily computed (Campbell et al., 2022) as:

$$q^d_{t|s} = P \exp\left(\Lambda \int_s^t \beta(\tau)d\tau\right)P^{-1} \tag{15}$$

In particular, we use a uniform stationary base rate $Q = \mathbf{1}\mathbf{1}^T - CI$, which is a natural choice for categorical discrete distributions as it admits a uniform distribution as its stationary distribution (Hoogeboom et al., 2021b; Austin et al., 2021). For simplicity of comparison we use the same schedule function as in Campbell et al. (2022) for all the experiments, and in Appendix C.1 we discuss other possible choices.

## 4.2 DISCRETE TIME SAMPLING FOR REVERSE PROCESS

Even given the learned conditional distribution $p_t(\cdot; \theta)$, simulating the reverse time process is much harder than simulating the forward process, as the reverse process cannot be factorized. For example, consider $x, y$ that are only different at the $d$-th site. The rate for the reversed process to jump to $y$ from $x$ at time $t$ is:

$$R^d_t(x_t, y; \theta) = \frac{p_t(X^d_t = y^d | x^{\backslash d}_t; \theta)}{p_t(X^d_t = x^d_t | x^{\backslash d}_t; \theta)} Q_t(y, x_t) \tag{16}$$

Such a jump rate depends on both the time $t$ and the value in the other dimensions $x^{\backslash d}_t$. Hence, we can not simulate each dimension in parallel, making exact simulation of the reverse time process $\overline{X}_t$ less efficient. Considering $R^d_t(x_t, y; \theta)$ is already an approximation of $R^d_t(x_t, y)$, we employ Euler's method for parallel simulation. Specifically, given $x_t$ at time $t$, we fix the rate in Equation 16, then determine the transition probabilities for dimension $d$ at time $t - \epsilon$ according to:

$$p^d_{t-\epsilon|t}(X^d_{t-\epsilon} = c | x^{\backslash d}_t; \theta) = \begin{cases} \epsilon R^d_t(x_t, X^d_{t-\epsilon} = c; \theta), & c \neq x^d_t \\ 1 - \epsilon \sum_{c' \neq x^d_t} R^d_t(x_t, X^d_{t-\epsilon} = c'; \theta), & c = x^d_t \end{cases} \tag{17}$$

We clip the quantities to ensure all probabilities are non-negative. Then, we collect a new value from each dimension to obtain a new state $y_{t-\epsilon}$, which has the factorized probability:

$$p_{t-\epsilon|t}(X_{t-\epsilon} = y_{t-\epsilon} | x_t; \theta) = \prod_{d=1}^D p^d_{t-\epsilon|t}(X^d_{t-\epsilon} = y^d_{t-\epsilon} | x^{\backslash d}_t; \theta) \tag{18}$$

In this way, we can update all dimensions of $x_t$ in parallel, making the simulation of the reverse time process much more efficient.

## 4.3 ANALYTICAL SAMPLING FOR REVERSE PROCESS

The Euler's method mentioned above assumes the rate matrix $R_\tau$ is fixed during $\tau \in [t - \epsilon, t]$. Such an approximation makes the samples $x_t$ deviate from the correct time slice marginal $q_t(\cdot)$, especially when the time step $\epsilon$ is large. To mitigate this approximation error, diffusion models usually introduce a corrector to improve sample quality (Song et al., 2020; Campbell et al., 2022), however this increases computational cost. Instead, we propose an alternative approach that leverages implicit modeling of $p_t(X^d | x^{\backslash d}_t; \theta)$. Specifically, for distribution $q$, we have:

$$q_t(X^d_t | x^{\backslash d}_t) = \sum_{x^d_0} q_{0|t}(x^d_0 | x^{\backslash d}_t) q_{t|0}(X^d_t | x^d_0) \tag{19}$$

Since $q_{t|0}$ is tractable, computing Equation 19 is efficient once we know $q_{0|t}(X^d_0 | x^{\backslash d}_t)$. Hence, we can replace the explicit approximation in Equation 12 by the following implicit approximation:

$$p_{0|t}(X^d_0 | x^{\backslash d}_t; \theta) \approx q_{0|t}(X^d_0 | x^{\backslash d}_t) \quad \Rightarrow \quad p_t(X^d_t | x^{\backslash d}_t; \theta) = \sum_{x^d_0} p_{0|t}(x^d_0 | x^{\backslash d}_t; \theta) q_{t|0}(X^d_t | x^d_0) \tag{20}$$

Equation 20 provides a tractable transformation from $p_{0|t}(X^d_0 | x^{\backslash d}_t; \theta)$ to $p_t(X^d_t | x^{\backslash d}_t; \theta)$. Hence, we can continue using the categorical ratio matching loss in Equation 14 to train $p_{0|t}(X^d_0 | x^{\backslash d}_t; \theta)$. To conduct backward sampling via this new parameterization, we consider the true reverse process:

$$q_{t-\epsilon|t}(X^d_{t-\epsilon} | x_t) = \sum_{x^d_0} q_{0|t}(x^d_0 | x_t) q_{t-\epsilon|0,t}(X^d_{t-\epsilon} | x^d_0, x_t) \tag{21}$$

$$= \sum_{x^d_0} \frac{q(x^d_t | x^d_0, x^{\backslash d}_t) q(x^d_0 | x^{\backslash d}_t)}{q(x^d_t | x^{\backslash d}_t)} \frac{q(x_t | x^d_0, X^d_{t-\epsilon}) q(X^d_{t-\epsilon} | x^d_0)}{q(x_t | x^d_0)} \tag{22}$$

$$\propto \sum_{x^d_0} q_{0|t}(x^d_0 | x^{\backslash d}_t) q_{t|t-\epsilon}(x^d_t | X^d_{t-\epsilon}) q_{t-\epsilon|0}(X^d_{t-\epsilon} | x^d_0) \tag{23}$$

By substituting $p_0(X_0^d|x_t^{\backslash d};\theta)$, we have:

$$p_{t-\epsilon|t}(X_{t-\epsilon}^d|x_t;\theta) \propto \sum_{x_0^d} p_{0|t}(x_0^d|x_t^{\backslash d};\theta) q_{t|t-\epsilon}(x_t^d|X_{t-\epsilon}^d) q_{t-\epsilon|0}(X_{t-\epsilon}^d|x_0^d) \qquad (24)$$

Thus, we obtain an analytical expression for the reverse process sampling that avoids the simulation error in Euler's method. Hence, we refer to this method as analytical sampling.

## 5 PARAMETERIZATION

The key to the simplicity of the objective function in Equation 14 is the special structure of the conditional marginal distributions $p_t(X_t^d|x_t^{\backslash d};\theta)$ and $p_{0|t}(X_0^d|x_t^{\backslash d};\theta)$ such that the marginals of $d$-th dimension do not depend on the current value $x_t^d$ at time $t$. However, this elegance in the objective brings additional challenges to design an efficient and flexible neural network parameterization. A key design constraint is that the predictions at a dimension $d$ must not depend on $x_t^d$, otherwise the information leak would make Equation 14 trivial to solve. The prediction can depend on the other coordinates in $x_t$ however. We offer three alternative architectures that are able to satisfy this constraint but incur different trade-offs between flexibility and computational cost. With out loss of generality we consider the parameterization of $p_t(X_t^d|x_t^{\backslash d};\theta)$, however the same designs can be directly applied to the parameterization of $p_{0|t}(X_0^d|x_t^{\backslash d};\theta)$.

### 5.1 ENERGY BASED MODELS

The most general and flexible form of parameterization is an Energy based model (EBM), where an arbitrary neural network $f_\theta(x,t): \mathcal{C}^D \times \mathbb{R} \mapsto \mathbb{R}$ can be used to specify the energy of any sample. In this case the conditional marginals can be modeled as:

$$p_t(X_t^d = c|x_t^{\backslash d};\theta) = \exp\left(-f_\theta([X_t^d = c, x_t^{\backslash d}], t)\right) / \sum_{c' \in \mathcal{C}} \exp\left(-f_\theta([X_t^d = c', x_t^{\backslash d}], t)\right) \quad (25)$$

where we overload the notation to use $[c, x_t^{\backslash d}]$ to denote $[x_t^0, x_t^1, \dots, X_t^d = c, x_t^{d+1}, \dots]$ such that only the $d$-th value is replaced with $c$. This modeling flexibility however comes at a high computational cost. Specifically, to evaluate $\prod_{d=1}^D p_t(X_t^d|x_t^{\backslash d};\theta)$ one needs $\mathcal{O}(D \times C)$ rounds of evaluation of $f_\theta$, which is computationally prohibitive when modeling high dimensional data with a deep neural parameterization of $f_\theta$. Nevertheless, this connection between EBMs and diffusion provides another way to learn time-dependent EBMs, which might be of independent interest.

### 5.2 MASKED MODELS

To alleviate the computation overhead of EBMs while preserving flexibility, a masked model is a natural choice. Specifically, let a masking function $m_d(x) = [x^1, \dots, x^{d-1}, \text{MASK}, x^{d+1}, \dots, x^D]$ replace the $d$-th dimension of a given $x$ to a special mask token MASK. Then one can formulate the following conditional parameterization:

$$p_t(X_t^d|x_t^{\backslash d};\theta) = \text{Softmax}\left(f_\theta(m(d), t)\right), \text{ where } f_\theta(x,t): \{\mathcal{C} \cup \text{MASK}\}^D \times \mathbb{R} \mapsto \mathbb{R}^C \quad (26)$$

Here $f$ can still be a general neural network with only a minor requirement on handling then masked token. Overall this approaches requires $\mathcal{O}(D)$ rounds of evaluation of $f_\theta$. Since we are dealing with discrete data, one can further reduce $D$ at the cost of increasing vocabulary size $C$, to further reduce the rounds of feed-forward evaluations of $f_\theta$.

### 5.3 HOLLOW TRANSFORMERS

Even though the two parameterizations above allow flexibility in the neural network design, they require a number of feed-forward evaluations that scales in the dimensionality and/or the vocabulary size of discrete data. As an alternative, we propose a Transformer variant that requires only $\mathcal{O}(1)$ feed-forward evaluations. The key constraint is that the diagonals of the Jacobian matrix of $p_t(X_t|x_t;\theta)$ be zero for any input $x_t$.

Many techniques have been developed to guarantee this property, including autoregressive masking (Germain et al., 2015; Vaswani et al., 2017), which creates a triangular Jacobian for multi-layer networks, or *hollow* masking (Chen & Duvenaud, 2019), which considers the full context but only permits a single layer of dense interaction between dimensions.

Here, to capture the full context for each dimension with expressive deep neural architectures, we introduce the *hollow Transformer*, a model that runs two autoregressive Transformers, one in each direction, for $L$-layers; see Figure 1. The hollow Jacobian matrix is obtained via the summation of upper and lower triangular Jacobians so the full context for each position is obtained. We add one additional Transformer layer at the top, where the *query* vector comes from the two embedding directions of the corresponding dimension, and attention on the *key* and *value* vectors are conducted jointly with the two directions. For clarity we omit details of each Transformer layer as we use the standard architecture (Vaswani et al., 2017). Overall, this specially designed architecture avoids any dependence on dimensionality or vocabulary size in terms of the number of feed-forward evaluations required, while being able to leverage the expressiveness of multi-layer Transformers.

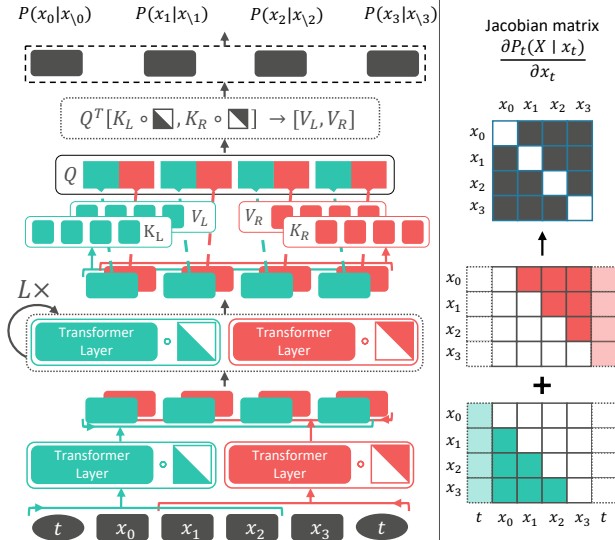

Figure 1: Diagram illustration of the Hollow Transformer.

## 6 EXPERIMENTS

We present an empirical evaluation of the proposed diffusion approach on synthetic data, CIFAR10 dataset and the monophonic music dataset. The primary goal is to compare the effectiveness of the proposed categorical ratio matching with the alternative parameterizations presented in Section 5. We use the time duration $T = 1$ and the uniform forward process in all cases. Experiments are conducted on machines equipped with TPU-v4 chips. For more details please refer to Appendix C.

### 6.1 DEEP EBMS ON SYNTHETIC DATA

As discussed in Section 5.1, one can leverage the EBMs for modeling conditional marginals. Here we verify this approach using a synthetic dataset from the discrete EBM learning community. Following prior works (Dai et al., 2020; Zhang et al., 2022), we use seven different distributions of 32-dimensional binary discrete data to evaluate different approaches. Each discrete distribution is converted from a 2-D continuous point $(x, y) \in \mathbb{R}^2$ by quantizing both $x$ and $y$ with 16-bit representations using a Gray code (Gray, 1953).

We parameterize the energy function $f_\theta(x, t)$ using the same 3-layer MLP as used in prior works (Dai et al., 2020; Zhang et al., 2022), with one minor change to directly add a sinusoidal embedding of $t$ into each hidden layer before activation. The uniform rate constant is set to 1.0 and we use a time resolution of $1e^{-3}$ for simulation. To measure sample quality, we follow (Zhang et al., 2022) and compare 4,000 generated samples to true data samples using exponential Hamming MMD (Gretton et al., 2012), repeated 10 times to report average results in Table 1. As baselines we include PCD (Tieleman, 2008), ALOE (Dai et al., 2020) with a larger network for dual parameterization, and EB-GFN (Zhang et al., 2022). Here we find that the proposed continuous-time categorical ratio matching approach is able to successfully learn a good EBM, yielding an MMD value that is consistently lower than the baseline methods. We also visualize the distribution obtained via SDDM in Figure 2 via reverse mapping of discrete Gray codes into 2-D points in continuous space. These plots show a similar distribution to the ground truth data (see Appendix C.2 for more information).

**Ablation study:** we further provide ablation studies on three different parameterizations proposed in Section 5 and two different sampling methods. For the full results please see Appendix C.2.1. Overall when 3-layer transformers are used in these three parameterizations, the performance are comparable to each other, which shows that the masked model and hollow model can achieve better quality-speed trade-offs than EBMs. We will revisit the comparison among samplers in next section.

### 6.2 IMAGE MODELING ON CIFAR10

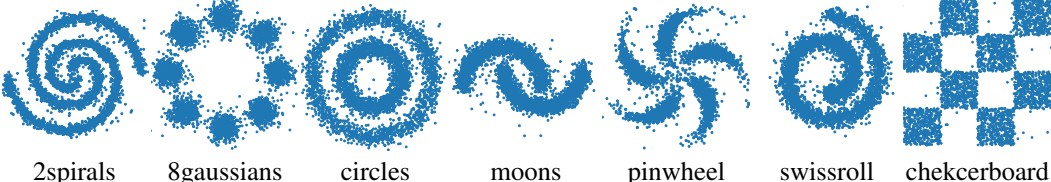

2spirals  8gaussians  circles  moons  pinwheel  swissroll  chekcerboard

Figure 2: Visualization of sampled discrete binary data in 2D space via decoding of Gray codes.

Table 1: Quality of generated binary samples from the learned EBMs, in terms of MMD with exponential Hamming kernel using bandwidth=0.1 (in units of $1 \times 10^{-4}$, the lower the better).

|  | 2spirals | 8gaussians | circles | moons | pinwheel | swissroll | checkerboard |
|---|---|---|---|---|---|---|---|
| PCD (Tieleman, 2008) | 2.160 | 0.954 | 0.188 | 0.962 | 0.505 | 1.382 | 2.831 |
| ALOE+ (Dai et al., 2020) | 0.149 | 0.078 | 0.636 | 0.516 | 1.746 | 0.718 | 12.138 |
| EB-GFN (Zhang et al., 2022) | 0.583 | 0.531 | 0.305 | **0.121** | 0.492 | 0.274 | 1.206 |
| SDDM (this paper) | **0.096** | **0.040** | **0.086** | 0.150 | **0.243** | **0.225** | **0.162** |

Next we evaluate image generation quality using the CIFAR10 dataset. In this case, each raw image has shape $(32, 32, 3)$ with 256 choices per each pixel value. We compare several diffusion models from the literature, with the main goal of understanding performance in the different spaces with either continu-

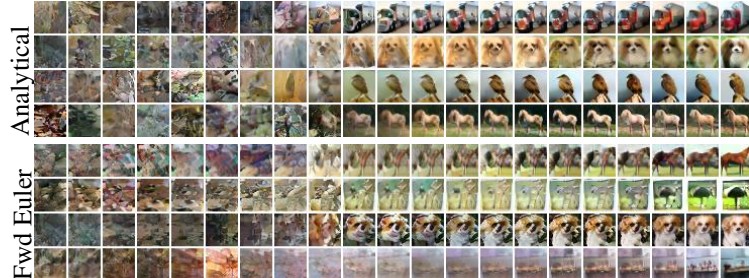

Figure 3: Visualization of reverse sampling with different samplers.

ous or discrete time formulations. We primarily evaluate the performance of the masked model formulation presented in Section 5.2, and report the commonly used Inception Score (IS) and Fréchet Inception Distance (FID) against the training set. As the image pixels are naturally discretized continuous values, we instead evaluate the proposed method in the quantization hash-code space via pretrained VQGAN (Esser et al., 2021). The main purpose of doing this is to 1) evaluate score matching on complex categorical discrete data; and 2) alleviate the burden of the Transformers modeling long sequences. The resulting data lies in a 64-dimensional categorical space with vocabulary size $|\mathcal{C}|$=512. Note that VQGAN is a lossy compressor where the reconstructed images from the decoder obtain IS=9.67 and FID=9.05, which is the upper-bound of categorical diffusion built on top of this vector quantizer (VQ). Here we simply use a 12-layer BERT-base architecture to parameterize the $f_\theta(x, t)$. See Appendix C.3 for more details about the VQ and architecture used.

From Table 2 we can see that, dealing with images in a categorical discrete space is generally much harder than an ordinal discrete space, as it loses ordinal structure as a prior. The proposed approach is able to approach the performance limit of VQ based categorical discrete diffusion, and improves the performance of D3PM and $\tau$LDR in the same VQ space with the same parameterization. In Figure 3 we visualize the reverse sampling procedure (unevenly sampled time steps) using two proposed samplers, where the analytical sampler achieves reasonable quality in fewer steps.

To quantitatively show why the continuous-time modeling would achieve better flexibility, we report the image quality with different number of sampling steps in Table 3. We can see as D3PM was trained with 1,000 steps, the performance drops quickly with fewer steps. Our continuous-time version can be more robust even with much fewer steps. Also the analytical sampler can be more robust when fewer steps are given, where the forward Euler may need corrector for better performance.

## 6.3 MONOPHONIC MUSIC MODELING

Finally, we conduct a study on discrete music generation following the same settings used in Campbell et al. (2022). This benchmark is originally from the Lakh pianoroll dataset (Raffel, 2016; Dong et al., 2018), cleaned by removing some trivial music sequences. In the end there are 6,000 music sequences for training and 973 sequences for evaluation. Each sequence is of length 256, where the

Table 2: Metrics on sample quality for different diffusion models on CIFAR10 dataset. Here Inception Score (IS) and Fréchet Inception Distance (FID) are compared. We follow the common practice to compare the 50,000 unconditionally sampled images from the model and the images from training dataset. Approaches with representations in different state spaces are listed in separate sections. *our re-implementation in VQ space, where we tuned configurations and report the best result.

| State space | Methods | IS ↑ | FID ↓ |
|---|---|---|---|
| Continuous state | DDPM (Ho et al., 2020) | 9.46 | 3.17 |
| | NCSN (Song et al., 2020) | 9.89 | 2.20 |
| Ordinal discrete state | D3PM Gauss (Austin et al., 2021) | 8.56 | 7.34 |
| | $\tau$LDR-0 (Campbell et al., 2022) | 8.74 | 8.10 |
| | $\tau$LDR-10 (Campbell et al., 2022) | 9.49 | 3.74 |
| Categorical discrete state | D3PM Uniform (Austin et al., 2021) | 5.99 | 51.27 |
| | D3PM Absorbing (Austin et al., 2021) | 6.78 | 30.97 |
| | VQGAN (Esser et al., 2021) reconstruction | 9.67 | 9.05 |
| | D3PM-VQ (Austin et al., 2021)* | 8.85 | 16.47 |
| | $\tau$LDR-VQ (Campbell et al., 2022)* | 5.71 | 40.06 |
| | SDDM-VQ (this paper) | 8.98 | 12.23 |

Table 3: FID/IS on CIFAR10 with different sampling steps, using continuous/discrete-time methods.

| FID ↓ with # steps | 250 | 200 | 100 | 50 | 40 | 20 | IS ↑ with # steps | 250 | 200 | 100 | 50 | 40 | 20 |
|---|---|---|---|---|---|---|---|---|---|---|---|---|---|
| D3PM | 32.61 | 38.56 | 62.15 | 84.72 | 90.1 | 102.3 | D3PM | 7.71 | 7.16 | 5.68 | 4.68 | 4.48 | 4.14 |
| SDDM (Euler) | **12.5** | **12.64** | 13.61 | 16.51 | 17.73 | 28 | SDDM (Euler) | 8.8 | 8.69 | 8.51 | 8.16 | 7.96 | 6.96 |
| SDDM (Analytical) | 13.41 | 13.39 | **13.29** | **14.99** | **15.96** | **21.06** | SDDM (Analytical) | **8.82** | **8.78** | **8.54** | **8.31** | **8.14** | **7.6** |

vocabulary size $|\mathcal{X}| = 129$ consists of 128 notes and a rest. The vocabulary is scrambled so there is no ordering information preserved. This creates a challenging categorical discrete data modeling task. Here we primarily compare against existing diffusion models in discrete spaces.

We use the simulation time $\epsilon=1e^{-3}$ for all methods. Since the dimensionality is high, the conditional marginals $p_{0|t}(X^d|x_t^{\backslash d})$ are parameterized by the proposed hollow Transformer (as discussed in Section 5.3) with 6 Transformer layers. Appendix C.4 provides more details

Table 4: Conditional music generation.

| Methods | Hellinger Distance | Proportion of Outliers |
|---|---|---|
| $\tau$LDR-0 Birth/Death | $0.3928 \pm 0.0010$ | $0.1316 \pm 0.0012$ |
| $\tau$LDR-0 Uniform | $0.3765 \pm 0.0013$ | $0.1106 \pm 0.0010$ |
| $\tau$LDR-2 Uniform | $0.3762 \pm 0.0015$ | $0.1091 \pm 0.0014$ |
| D3PM Uniform | $0.3839 \pm 0.0002$ | $0.1137 \pm 0.0010$ |
| SDDM (this paper) | $0.3736 \pm 0.0024$ | $0.1093 \pm 0.0016$ |

about the training and parameterizations. The evaluation was conducted on the held-out test set, where for each sequence, a prefix of length 32 is given as conditioning and the models are asked to generate the remaining 224 tokens. We follow the same evaluation protocol as in Campbell et al. (2022) and report the Hellinger Distance and Proportion of Outliers after 5 runs per each test case in Table 4. Overall we can see the proposed SDDM is able to consistently improve the two metrics.

## 7 LIMITATION AND CONCLUSION

In this paper we presented a new learning and sampling paradigm for continuous-time diffusion models in categorical discrete spaces. We developed an extension of score matching to categorical discrete variables, showing that the corresponding continuous-time score matching properly aligns the reverse process with the posterior of the forward processes. The new learning paradigm also naturally introduces new sampling algorithms. Despite the promising preliminary results, there are still limitations in the current treatment. The main bottleneck is the design of the conditional marginal parameterization, which requires non-trivial trade-offs between computational cost and flexibility of the architectures; score matching for general categorical discrete variables does not benefit from prior knowledge about ordinal discrete data; and finally unifying score matching between continuous and discrete spaces would be needed to handle data in mixed spaces. We believe this initiative sheds new light on score matching for discrete-space diffusion.

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

## A RELATED WORK

The discrete diffusion model has actually been described in the original paper by (Sohl-Dickstein et al., 2015), in which a binomial diffusion process is considered for binary variable modeling. After the resurgence of the diffusion models with successful applications on continuous variable distribution modeling (Ho et al., 2020; Song et al., 2020), the recent discrete diffusion extensions mainly focus on the design of forward corruption kernels. The multinomial diffusion extension is proposed (Hoogeboom et al., 2021b), which inspires more structured perturbation operations in the forward process in Austin et al. (2021) beyond uniform noise injection, including discretized Gaussian perturbation, transition proportion to token embedding distance, and absorbing state corruption. Furthermore, Hoogeboom et al. (2021a) introduce the standard masking operation in language model into forward process, and Johnson et al. (2021) consider the injection and deletion operation for corruption that is beyond the in-place perturbation. Discrete diffusion models demonstrate better flexibility and have been used as alternative of autoregressive models, *e.g.*, in Gu et al. (2022); Cohen et al. (2022), discrete diffusion models are used for latent codes quantization and composed with decoder/encoder in VQ-VAE. These models still follow the *finite-step* corruption in forward processes and exploit MLE-related objectives for learning, which is orthogonal to our focus.

The most related work is Campbell et al. (2022), which generalize the discrete diffusion processes by characterizing the continuum of discrete variable evolving through continuous-time Markov chain. Our work also considers the continuous-time discrete diffusion. The major differences lie in two-fold: **i)**, we designed score-based learning, while the learning in Campbell et al. (2022) relies on ELBO **ii)**, our derivation admits an analytical sampling strategy during backward sampling, in addition to the commonly used numerical simulations in Campbell et al. (2022). Together with (Campbell et al., 2022), we complete the missing puzzle of discrete diffusion models in continuous-time framework.

## B DEFFERED PROOF

### B.1 PROOF FOR PROPOSITION 3.1

*Proof.* For a time $u \in (0, T)$, denote the filtration $\mathcal{F}_u = \sigma\{X_v : v \in [0, u]\}$, which is the sigma algebra of the events, $X_u$, from time 0 to time $u$. Also, denote the filtration for the reverse time process as $\overline{\mathcal{F}}_u = \sigma\{\overline{X}_v : v \in [0, u]\} = \sigma\{X_v : v \in [u, T]\}$, which is the sigma algebra of events, $X_u$, from time $u$ to time $T$. By the Markov property, $\mathcal{F}_u$ and $\overline{\mathcal{F}}(u)$ are conditionally independent given $X_u$. Now, consider any $A \in \overline{\mathcal{F}}_{T-t}$, and any $s < t$. Then we have

$$\mathbb{P}(\overline{X}_{T-s} = x_s | \overline{X}_{T-t} = x_t, A) \tag{27}$$

$$= \mathbb{P}(X_s = x_s | X_t = x_t, A) = \frac{\mathbb{P}(X_s = x_s, X_t = x_t, A)}{\mathbb{P}(X_t = x_t, A)} \tag{28}$$

$$= \frac{\mathbb{P}(A | X_s = x_s, X_t = x_t)\mathbb{P}(X_t = x_t | X_s = x_s)\mathbb{P}(X_s = x_s)}{\mathbb{P}(A | X_t = x_t)\mathbb{P}(X_t = x_t)} \tag{29}$$

$$= \mathbb{P}(X_t = x_t | X_s = x_s)\frac{\mathbb{P}(X_s = x_s)}{\mathbb{P}(X_t = x_t)} = q_{t|s}(x_t|x_s)\frac{q_s(x_s)}{q_t(x_t)} \tag{30}$$

is independent of $A$, which proves the reverse time process $\overline{X}_t$ is a continuous time Markov chain. Also, from the derivation above we have:

$$q_{t|s}(x_t|x_s)\frac{q_s(x_s)}{q_t(x_t)} = \mathbb{P}(X_s = x_s | X_t = x_t, A) = \mathbb{P}(X_s = x_s | X_t = x_t) = q_{s|t}(x_s|x_t) \tag{31}$$

Hence, we have established the proposition. □

## B.2 PROOF FOR PROPOSITION 3.2

*Proof.* Since the transition kernel for the time reversal process $\overline{X}_t$ satisfies Equation 9, we consider its time derivative. For $x \neq y$, we have

$$\frac{d}{dt} q_{T-t|T-s}(y|x) = \frac{d}{dt} \left[ \frac{q_{T-t}(y)}{q_{T-s}(x)} q_{T-s|T-t}(x|y) \right] \tag{32}$$

$$= \frac{\frac{d}{dt} q_{T-t}(y)}{q_{T-s}(x)} q_{T-s|T-t}(x|y) + \frac{q_{T-t}(y)}{q_{T-s}(x)} \frac{d}{dt} q_{T-s|T-t}(x|y) \tag{33}$$

For the first term in Equation 33, we use Kolmogorov forward equation:

$$\frac{d}{dt} q_{T-t}(y) = \frac{d}{dt} \sum_{x_0 \in \mathcal{X}} \pi_{\text{data}}(x_0) q_{T-t|0}(y|x_0) \tag{34}$$

$$= \sum_{x_0 \in \mathcal{X}} \pi_{\text{data}}(x_0) \frac{d}{dt} q_{T-t|0}(y|x_0) \tag{35}$$

$$= \sum_{x_0 \in \mathcal{X}} \pi_{\text{data}}(x_0) \sum_z -q_{T-t|0}(z|x_0) Q_{T-t}(z,y) \tag{36}$$

$$= -\sum_z q_{T-t}(z) Q_{T-t}(z,y) \tag{37}$$

Since $\mathcal{X}$ is a finite space, the summation over $z$ is finite, hence we obtain:

$$\lim_{t \to s} \frac{\frac{d}{dt} q_{T-t}(y)}{q_{T-s}(x)} q_{T-s|T-t}(x|y) = \lim_{t \to s} \frac{-\sum_z q_{T-t}(z) Q_{T-t}(z,y)}{q_{T-s}(x)} q_{T-s|T-t}(x|y) = 0 \tag{38}$$

For the second term in Equation 33, we use Kolmogorov backward equation to obtain the derivative:

$$\frac{d}{dt} q_{T-s|T-t}(x|y) = \sum_z Q_{T-s}(z,x) q_{T-s|T-t}(y,z) \tag{39}$$

Then we have:

$$\lim_{t \to s} \frac{q_{T-t}(y)}{q_{T-s}(x)} \frac{d}{dt} q_{T-s|T-t}(x|y) = \frac{q_{T-s}(y)}{q_{T-s}(x)} Q_{T-s}(y,x) \tag{40}$$

By combining these two results, and using the property that:

$$R_{T-s}(x,y) = \lim_{t \to s} \frac{d}{dt} q_{T-t|T-s}(x|y) = 0 + \frac{q_{T-s}(x)}{q_{T-s}(y)} Q_{T-s}(x,y) \tag{41}$$

then relabelling $T-s$ by $t$ yields:

$$R_t(x,y) = \frac{q_t(y)}{q_t(x)} Q_t(x,y) \tag{42}$$

$\square$

## B.3 PROOF FOR PROPOSITION 3.3

*Proof.* For one direction, if $\pi_1 = \pi_2$, it is trivial that their conditional distributions match. For the other direction, consider $x, y \in \mathcal{X}$ and an arbitrary probability distribution $\pi$. We have:

$$\frac{\pi(y)}{\pi(x)} = \frac{\pi(y^1, y^2, y^3, ..., y^{D-1}, y^D)}{\pi(x^1, x^2, x^3, ..., x^{D-1}, x^D)} \tag{43}$$

$$= \frac{\pi(y^1, y^2, y^3, ..., y^{D-1}, y^D)}{\pi(x^1, y^2, y^3, ..., y^{D-1}, y^D)} \frac{\pi(x^1, y^2, y^3, ..., y^{D-1}, y^D)}{\pi(x^1, x^2, y^3, ..., y^{D-1}, y^D)} \cdots \frac{\pi(x^1, x^2, ..., x^{D-1}, y^D)}{\pi(x^1, x^2, ..., x^{D-1}, x^D)} \tag{44}$$

$$= \prod_{d=1}^{D} \frac{\pi(y^d | x^{1:d-1}, y^{d+1:D})}{\pi(x^d | x^{1:d-1}, y^{d+1:D})} \tag{45}$$

Thus the probability ratio can be decomposed as a product completely determined by the singleton conditional distributions. Hence, if $\pi_1$ and $\pi_2$ have the same singleton conditional distributions, then $\pi_1(y)/\pi_1(x) = \pi_2(y)/\pi_2(x), \forall x, y$. As they are both distributions, we can easily see $1/\pi_1(x) = \sum_y \pi_1(y)/\pi_1(x) = \sum_y \pi_2(y)/\pi_2(x) = 1/\pi_2(x), \forall x$. Hence $\pi_1 = \pi_2$. $\square$

### B.4 Proof for Proposition 3.4

The key idea for the proof is that the conditional distribution on $d$-th dimension does not rely on the value $x^d$. That is to say, the value of $q_t(X^d = c|x_t^{\backslash d}) \log p_t(X^d = c|x_t^{\backslash d}; \theta)$ does not depend on $x_t^d$. Hence, we have:

$$\sum_{x_t \in \mathcal{X}} q_t(x) \sum_{d=1}^{D} \sum_{c \in \mathcal{C}} q_t(X^d = c|x_t^{\backslash d}) \log p_t(X^d = c|x_t^{\backslash d}; \theta) \tag{46}$$

$$= \sum_{d=1}^{D} \sum_{c \in \mathcal{C}} \sum_{x_t \in \mathcal{X}} q_t(x_t) \frac{q_t(x_t^{\backslash d}, X^d = c)}{\sum_{c' \in \mathcal{C}} q_t(x_t^{\backslash d}, X^d = c')} \log p_t(X^d = c|x_t^{\backslash d}; \theta) \tag{47}$$

$$= \sum_{d=1}^{D} \sum_{c \in \mathcal{C}} \sum_{x_t^{\backslash d}} \sum_{c'' \in \mathcal{C}} q_t(x_t^{\backslash d}, c'') \frac{q_t(x_t^{\backslash d}, X^d = c)}{\sum_{c' \in \mathcal{C}} q_t(x_t^{\backslash d}, X^d = c')} \log p_t(X^d = c|x_t^{\backslash d}; \theta) \tag{48}$$

$$= \sum_{d=1}^{D} \sum_{c \in \mathcal{C}} \sum_{x_t^{\backslash d}} \frac{q_t(x_t^{\backslash d}, X^d = c)}{\sum_{c' \in \mathcal{C}} q_t(x_t^{\backslash d}, X^d = c')} \log p_t(X^d = c|x_t^{\backslash d}; \theta) \sum_{c'' \in \mathcal{C}} q_t(x_t^{\backslash d}, c'') \tag{49}$$

$$= \sum_{d=1}^{D} \sum_{c \in \mathcal{C}} \sum_{x_t^{\backslash d}} q_t(x_t^{\backslash d}, X^d = c) \log p_t(X^d = c|x_t^{\backslash d}; \theta) \tag{50}$$

$$= \sum_{d=1}^{D} \sum_{x_t \in \mathcal{X}} q_t(x_t) \log p_t(X^d = x_t^d|x_t^{\backslash d}; \theta) \tag{51}$$

$$= \sum_{x_t \in \mathcal{X}} \sum_{d=1}^{D} q_t(x_t) \log p_t(X^d = x_t^d|x_t^{\backslash d}; \theta) \tag{52}$$

By substituting this result into original score matching loss function:

$$\theta^* = \arg \min_{\theta} \int_0^T \sum_{x_t \in \mathcal{X}} q_t(x_t) \left[ \sum_{d=1}^{D} \left( -\sum_{c \in \mathcal{C}} q_t(X^d = c|x_t^{\backslash d}) \log p_t(X^d = c|x_t^{\backslash d}; \theta) \right) \right] dt \tag{53}$$

we obtain a simplified and tractable loss function:

$$\theta^* = \arg \min_{\theta} \int_0^T \sum_{x_t \in \mathcal{X}} q_t(x_t) \left[ \sum_{d=1}^{D} -\log p_t(X^d = x_t^d|x_t^{\backslash d}; \theta) \right] dt \tag{54}$$

and prove the proposition 3.4.

## C Experimental details

### C.1 Noise schedule of $\beta(t)$

Empirically we find the following time schedule $\beta(t)$ to be effective in some situations:

$$\int_0^t \beta(\tau)d\tau = -\left( \cos \frac{\pi}{2} t \right)^{\frac{1}{2}} + 1 \tag{55}$$

This provides a cosine style noise schedule so that the forward process has a reasonable noise level to contribute to sample quality (Nichol & Dhariwal, 2021).

### C.2 Synthetic data

To parameterize $f_\theta(x, t)$, we use the same 3-layer MLP as used in Zhang et al. (2022). Each hidden layer has dimensionality of 256, with elu activations. We use a constant learning rate of $1e^{-4}$ and train the model using 300k steps, where the per-step batch size is 128. The data is generated on the fly, using the data generator provided by Dai et al. (2020).

Table 5: Conditional music generation compared against ground truth completions, with different noise schedules.

| Methods | Hellinger Distance | Proportion of Outliers |
|---|---|---|
| SDDM | $0.3736 \pm 0.0024$ | $0.1093 \pm 0.0016$ |
| SDDM with (55) | $0.3735 \pm 0.0025$ | $0.1071 \pm 0.0017$ |

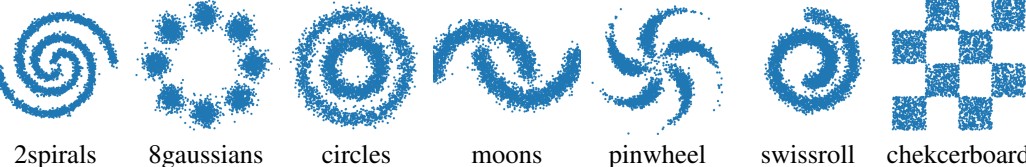

| 2spirals | 8gaussians | circles | moons | pinwheel | swissroll | chekcerboard |
|---|---|---|---|---|---|---|

Figure 4: Visualization of the true data in 2D space via decoding of Gray codes.

### C.2.1 ABLATION STUDY

In this section we provide the ablation study on three parameterization methods proposed in section 5, as well as the two proposed sampling methods (namely the forward Euler's method, denoted as Fwd Euler, and the analytical sampling method, denoted as analytical). Overall we can see all the combinations of parameterization+sampler achieve reasonable results. The comparison among different samplers and models are mixed, which indicates that more efficient parameterizations like masked or hollow models can achieve better quality-speed trade-offs than EBM in this synthetic data.

### C.3 EXPERIMENTS ON CIFAR10

**Vector Quantization**   We model the images in a learned categorical discrete latent space with a Vector Quantized Variational Autoencoder (VQ-VAE) (Van Den Oord et al. (2017)). The VQVAE encoder maps an image in $H \times W \times 3$ to $\frac{H}{4} \times \frac{W}{4}$ tokens with a vocabulary size of 512. During training, we use the VQGAN (Esser et al. (2021)) variant which uses a GAN loss and a perceptual loss in addition to the ELBO objective.

We follow the implementation of VQGAN in MaskGIT (Chang et al. (2022)) with adaptations for a smaller down sample rate and fewer parameters. In particular, we use three convolution blocks in the encoder, with average pooling for down sampling between them. The three blocks use 64, 128, and 256 filters respectively, where each block consists of two standard residual blocks. We use nearest neighbor lookup to map the encoder outputs to a token index, according to a codebook of $512 \times 256$. The decoder mirrors the encoder architecture. The overall parameters for the VQVAE is 12.35M.

We use the straight-through gradient estimator (Van Den Oord et al. (2017)) for the quantization part during training. To acquire a general VQ-VAE model as the bridge between pixel space and latent space, we train it on the ImageNet dataset (Deng et al. (2009)) at $64 \times 64$ resolution for 90 epochs. The model is trained with the Adam optimizer ($\beta_1 = 0, \beta_2 = 0.99$) (Kingma & Ba (2014)), using a peak learning rate of $1e - 4$ with a schedule of linear warm up and cosine decay. The GAN loss and perceptual loss are added with weight 0.1.

For the $32 \times 32$ images in the CIFAR10 dataset, $8 \times 8$ latent codes are produced. The reconstruction achieves FID of 9.05 and IS of 9.67.

**Paramterization and training**   We parameterize $f_\theta(x, t)$ using the masked modeling (see section 5.2). The backbone of the neural network is the same as BERT-base, which consists of 12 layers of Transformers, where each layer has 12 attention heads, embedding size of 768 and hidden layer size of 3072 for MLPs. We simply concatenate the time embedding representation of $t$ together with the other tokens. After obtaining the final embedding of each token, we feed that into a 2-block ResNet (using the same parameterization as used in Campbell et al. (2022) for their music

Table 6: Ablation study of SDDM on synthetic dataset, using the same experimetal protocol as in 1

| Methods | 2spirals | 8gaussians | circles | moons | pinwheel | swissroll | chekcerboard |
|---|---|---|---|---|---|---|---|
| EBM+Analytical | 0.250 | 0.075 | 0.267 | 0.037 | 0.323 | 0.305 | 0.383 |
| Masked+Analytical | 0.143 | 0.080 | 0.126 | 0.064 | 0.242 | 0.116 | 0.345 |
| Hollow+Analytical | 0.227 | 0.052 | 0.229 | 0.117 | 0.049 | 0.201 | 0.557 |
| EBM+Fwd Euler | 0.199 | 0.022 | 0.173 | 0.095 | 0.167 | 0.287 | 0.211 |
| Masked+Fwd Euler | 0.120 | 0.028 | 0.204 | 0.195 | 0.064 | 0.086 | 0.248 |
| Hollow+Fwd Euler | 0.179 | 0.088 | 0.077 | 0.034 | 0.288 | 0.281 | 0.148 |

generation experiments) and predict logits for the masked position. We use a constant uniform rate of 0.007 for the forward process.

We train the model with 4x4x4 TPU-v4 chips, with batch size of 128. The training is done after 700k steps in about 20h. The learning rate is warmed up from 0 to $1e^{-4}$ during the first 3% steps of training, and then decays to 0 in a linear schedule. The final evaluation is done on the exponential moving average of the model parameters, with the decay rate of 0.999.

## C.4    MUSIC DATASET

We parameterize the $f_\theta(x, t)$ using the proposed hollow Transformer (see section 5.3). Each Transformer component has 6 layers with embedding size of 256, 8 attention heads and hidden dimension of 2048. We simply concatenate the time embedding together with other tokens for the feed-forward. We experiment it using 2x2 TPU-v4 chips, and it took around 12 hours to get to convergence, or roughly 2m steps with batch size of 64.

