# OpenReview forum: "Score-based Continuous-time Discrete Diffusion Models"
_ICLR.cc/2023/Conference — ICLR 2023 poster_

### Official Review · Reviewer_tzeK · 2022-10-20

**Confidence:** 4
**Correctness:** 3
**Technical Novelty And Significance:** 3
**Empirical Novelty And Significance:** Not applicable
**Recommendation:** 6

**Clarity, Quality, Novelty And Reproducibility:**

The extension of the ratio test for binary variables proposed in Hyvarinen (2007) to the categorical counterpart is straightforward but make it working with the proposed model is a non-trivial task and requires a lot of architectural designs. Unfortunately, the presentation of this paper makes it difficult to understand. Besides issues mentioned above, a few more:
- In Section 6.3, it states that "we use the simulation time $\tau=1e^{-3}$ for all methods". Is this $\tau$ the same as the $\epsilon$ used in Section 4,2 and 4.3? Or is this something not referenced previously? Which simulation time is used for the CIFAR10 data?
- In Appendix B.2, how does equation (34) equal to equation (35)? is this a typo?
- Do we have the similar SDE formulations as in equation (5,6) for the discrete data since we already have the score function? Is it just a simple replacement of the proposed score function?

For the reproducibility, there are certain experimental details presented in the main text and appendix, but I think it is still not quite easy to reproduce the results presented in Section 6. Hopefully the authors would release their code.

**Strength And Weaknesses:**

*Strengths*

I think this work studies an interesting problem and it is nice to see the authors take the effort to extend the score functions to the categorical data and manage to leverage it to derive a score-matching based objective for estimating the continuous-time diffusion models. The main idea of using score-based objectives instead of ELBO surrogates is well-motivated, and the related works are properly referenced.

The authors have tried many different parametrizations and a new design for the conditional marginals, and it is also nice to see the proposed score-based learning naturally introduces a new analytic sampling algorithm. Experiments with toy 2-D datasets, CIFAR-10, and monophonic music dataset show the soundness of the proposed method, and the results are competitive with previous state-of-the-art methods.


*Weaknesses*

The presentation is on the weaker side, and I think the paper can be significantly improved by having better clarity.
- The authors extend the ratio test proposed in Hyvarinen (2007) to the categorical variables but point to equation (14) as the "score-matching" objective. However, it does not resemble those used in Hyvarinen (2007) or Song et al. (2020). It would be better if the authors could discuss the relationship of the proposed objective to the classic ones.
- It would be nicer if the authors could discuss more regarding the similarities and differences from Campbell et al. (2022). For example, is there any subtle difference between the derivation of the continuous time modeling in Section 3.1 and those in Section 3.1 in Campbell et al. (2022)? Also, comparisons between the learning objectives would also make it easier to understand the benefits of the proposed method.

The parametrization of the conditional marginal distributions. Even though the authors have tried to design an efficient structure through different architecture choices including the hollow transformers, the current design is more computationally demanding comparing to existing methods and prevents it from directly modeling the CIFAR10 data with resolution as small as 32x32, not to mention other datasets with higher resolution. So that it is not easy to perform an apple-to-apple comparison with existing work.

The experimental validations are not very convincing. Two benefits of the proposed method are discussed in the introduction, including 1) the continuous-time discrete diffusion formulation is more flexible comparing to the discrete-time counterparts and may admit more effective estimation and generation; 2) the score-based learning yields superior estimation quality comparing to the ELBO surrogate used in Campbell et al. (2022) for the continuous-time discrete diffusion models. However, neither of them is convincingly validated in the experiment section. First, there is no formal comparison of the computations used during training and generation between the proposed model and the discrete-time ones. Second, the noise schedule adopted here is different from what used in Campbell at al. (2022). It has been shown that noise schedule can have noticeable impact on the final performance. Thus, this difference may question the validity of the conclusion that the better performance of the proposed model can be solely attributed to the score-based learning process. Third, it would be desirable to perform some ablation studies to validate the designs. For example, comparisons among all three parametrizations in the synthetic experiments and comparisons between the sampling methods discussed in Section 4.2 and 4.3.

**Summary Of The Paper:**

The work examines the problem of continuous-time discrete diffusion models, and the paper mainly focuses on 1) extending the existing discrete-time discrete diffusion models to the continuous-time via stochastic jump processes, 2) extending the score function to generic categorical variables and deriving a score-matching based learning for the model, 3) different sampling methods for the reverse process. Experimental validations on synthetic data, image and music generation have shown promising results comparing to existing discrete-time and continuous-time discrete diffusion models.

**Summary Of The Review:**

I think the score-based continuous-time extension of the discrete diffusion models is interesting and the paper makes a novel contribution to it. However, I am leaning towards the negative side due to the weak presentation and not-so-convincing experiments.

---

> ### Author Response · Authors · 2022-11-19
> **Response to Reviewer tzeK (Part I)**
>
> We thank the reviewer for providing helpful feedback! Please see our response below:
>
> ### Relation to Hyvarinen (2007) and Song et al. (2020)
>
>
> As the reviewer pointed out, Hyvarinen (2007) generalizes the score matching from continuous variables to binary random variables. Song et al. (2020) define the score matching model for diffusion process in continuous model.
> In this work, out contribution lies in 1) generalizing Hyvarinen (2007) by extending their binary ratio matching to **categorical** random variables, 2) generalizing Song et al. (2020) by extending the diffusion process to discrete space via Markov jump process (continuous time Markov chain) and using categorical score matching to learn the categorical score function.
> In our paper we use the ratio matching or categorical score matching interchangeably, as it is a categorical generalization of binary ratio matching from Hyvarinen (2007). We have refined the paper accordingly to make the definition clear, hope this would resolve the potential confusion, but we are also open to further suggestions from you!
>
> ### Comparison with Campbell et al. (2022)
>
> We both follow the standard definition of Continuous-time Markov chain (CTMC), so the derived continuous time models in the forward process are mathematically equivalent, so are the time reversal models.
> One of the major differences is the learning framework. The framework of Campbell et al. (2022) is derived from the minimization of KL divergence $D_\text{KL}(p_0(\cdot; \theta \| q_0(\cdot))$, and the ELBO is used as a surrogate in practice. Our framework is derived from the score matching (or categorical ratio matching) view, which directly fits the rate in the time reversal CTMC.
> Under the new framework, we derived two new simulation methods (namely the forward Euler and analytical sampling in the paper) that are suitable for categorical discrete distributions.
>
>
> ### “...perform an apple-to-apple comparison with existing work…the score-based learning yields superior estimation quality compared to the ELBO surrogate used in [1]...”
>
> Following reviewer’s suggestion, we re-implemented both D3PM [2] and $\tau$-LDR [1] to run in the **same** vector quantization (VQ) space as our SDDM, and report the results on CIFAR10 image modeling to obtain the apple-to-apple comparison.
>
> | Methods | IS $\uparrow$ | FID $\downarrow$ |
> |---|---|---|
> | D3PM* | 8.85 | 16.47 |
> | \tau-LDR* | 5.71 | 40.06 |
> | SDDM | **8.98** | **12.23** |
>
>
> We can see when all the methods are compared in the **same architecture and same categorical discrete space**, our SDDM outperforms others in terms of image generation quality. With that said, and as we pointed out in the limitation/conclusion section, our current method is not efficient enough in handling high-dimensional images, and we hope to improve it in future works.
>
> > *We re-implemented both baselines to run in the same VQ space with the same architectures as the proposed SDDM. We tried our best to tune $\tau$-LDR with different learning rate schedules, different approximations (one-pass/two-passes as denoted in their paper).
>
> ### “... the continuous-time discrete diffusion formulation is more flexible comparing to the discrete-time counterparts…”
>
> The major difference between continuous and discrete time models is that the continuous time model admits flexible discretization during inference/sampling, which allows better control over the generation quality and computation cost. To see why this is the case, we report the FID (IS shows similar results) with different numbers of sampling steps, in the VQ space.
>
> | Samplers | 1000 | 250 | 200 | 100 | 50 | 40 | 20 |
> |---|---|---|---|---|---|---|---|
> | D3PM [2] | 16.47 |  32.61 | 38.56 | 62.15 | 84.72 | 90.1 | 102.3 |
> | SDDM | 12.00 | 12.5 | 12.64 | 13.61 | 16.51 | 17.73 | 28 |
>
> Overall we can see the performance of D3PM drops a lot when fewer numbers of steps are used during inference, demonstrating our claim about the benefits of continuous-time diffusion in inference. One potential reason is that D3PM fixes the number of steps (in this case 1000) during training, while for continuous-time models the discretization is not introduced yet during training, and thus admits the flexibility during backward sampling.
>
> **References:**
> >[1] A Continuous Time Framework for Discrete Denoising Models, Campbell et.al, 2022.\
> [2] Structured Denoising Diffusion Models in Discrete State-Spaces, Austin et.al, 2021.

---

> > ### Comment · Reviewer_tzeK · 2022-12-06
> > **Response and raising my score**
> >
> > I want to thank the authors for the detailed response and efforts to make the quantitative analysis more robust. These have addressed most of my concerns especially around the comparisons with baselines and the flexibility of continuous-time formulation. I also appreciate that the authors have explicitly listed the limitation that the current method is not efficient enough in handling high-dimensional data.

---

> > > ### Author Response · Authors · 2022-12-08
> > > **thank you!**
> > >
> > > We thank the reviewer for the valuable feedbacks which are indeed helpful! We also appreciate reviewer's effort during the review process.

---

> ### Author Response · Authors · 2022-11-19
> **Response to Reviewer tzeK (Part II)**
>
> ### “...more computationally demanding comparing to existing methods…First, formal comparison of the computations used during training and generation…”
>
> Regarding the computations, both [1, 2] and our SDDM should have the same asymptotic computational complexity (in terms of evaluations of $p_{0|t}$) during training/inference. However in practice it would be affected by the number of sampling steps and the computation cost of parameterization. Regarding these two aspects:
> 1) From the above comparison we can see our SDDM requires much fewer steps than D3PM to achieve the same quality.
> 2) We agree with the reviewer that the specific masked-model parameterization we used is more expensive, and indeed we listed this as one of the limitations in Section 7. However, one good news is that this parameterization can be embarrassingly parallelized with more devices. On the contrary the sampling steps need to happen sequentially, and cannot benefit from more devices. We have added such discussion in our revision.
>
>
>
> ### “Second, the noise schedule adopted here is different…”
>
> We found the new noise schedule gives minor improvements but creates unnecessary distractions to the paper. Following reviewer’s suggestion, we have rerun all the experiments using the same forward process as in $\tau$-LDR [1]. We updated all the tables correspondingly in the paper, and moved the ablation study on the noise schedule into the appendix C1. Overall our proposed method still obtains comparable or better results. We highlighted the updated numbers in the paper with blue color.
>
> ### “Third, it would be desirable to perform some ablation studies … all three parametrizations… between the sampling methods…”
>
> Thank you for your constructive feedback! We have followed your suggestions to provide ablation studies and revised the paper accordingly. We include the results below for your convenience.
>
> **Ablation study on synthetic datasets**
>
> We first show the results on synthetic data. We use the same MMD evaluation protocol as in Table 1 in the main paper. All three parameterizations use a 3-layer transformer with embedding size 64  and MLP hidden layer size 256. For each method, we try both proposed sampling methods, namely the forward Euler’s method (Fwd Euler) and the analytical sampling method (Analytical). Note that the results of EBM reported in Table 1 are with 3-layer MLP with the purpose of comparing against other EBM training methods, which is different from the results with transformers below.
>
> | Model+Sampler | 2spirals | 8gaussians | circles | moons | pinwheel | swissroll | checkerboard|
> |---|---|---|---|---|---|---|---|
> | EBM+Analytical | 0.250| 0.075 | 0.267| 0.037| 0.323| 0.305| 0.383|
> | Masked+Analytical | 0.1431 | 0.080 | 0.126 | 0.064 | 0.242 | 0.116 | 0.345|
> | Hollow+Analytical | 0.227 | 0.052 | 0.229 | 0.117 | 0.049 | 0.201 | 0.557 |
> | EBM+Fwd Euler | 0.199 | 0.022 | 0.173 | 0.095 | 0.167 | 0.287 | 0.211 |
> | Masked+Fwd Euler | 0.120 | 0.028 | 0.204 | 0.195 | 0.064 | 0.086 | 0.248 |
> | Hollow+Fwd Euler | 0.179 | 0.088 | 0.077 | 0.034 | 0.288 | 0.281 | 0.148 |
>
> Overall we can see all the combinations of parameterization+sampler achieve reasonable results. The comparison among different samplers and models are mixed, which indicates that more efficient parameterizations like masked or hollow models can achieve better quality-speed trade-offs than EBM in this synthetic data.
>
> **Ablation study on CIFAR10 dataset**
>
> As different sampling methods show similar results on synthetic data, here we take further study to compare the sampled image qualities using different samplers on the same trained model (the one used in the main paper). As asymptotically these samplers can be equivalent in simulating the processes, it would be more interesting when fewer steps are used. To save space we include the table of FID (the results on IS are also consistent) below and for full results please see our updated Section 6.2.
>
> | Samplers | 250 | 200 | 100 | 50 | 40 | 20 |
> |---|---|---|---|---|---|---|
> | Fwd Euler | 12.5 | 12.64 | 13.61 | 16.51 | 17.73 | 28 |
> | Analytical | 13.41 | 13.39 | 13.29 | 14.99 | 15.96 | 21.06 |
>
> We can see when the number of steps are small, the analytical sampler archives relatively better results. In this case the forward Euler may need correctors to guide the sampling.
> When more steps are used, the forward Euler simulator can be more stable.
>
> **References:**
> >[1] A Continuous Time Framework for Discrete Denoising Models, Campbell et.al, 2022.\
> [2] Structured Denoising Diffusion Models in Discrete State-Spaces, Austin et.al, 2021.

---

> ### Author Response · Authors · 2022-11-19
> **Response to Reviewer tzeK (Part III)**
>
> ### “In Section 6.3…Is this τ the same as the ϵ… Which simulation time is used for CIFAR10”
>
> Yes, the $\tau$ is the same as the $\epsilon$ in section 4.2 and 4.3. We thank the reviewer for pointing out our inconsistency in notation. We have fixed $\tau$ to $\epsilon$ in our revised version. For CIFAR10 we set simulation time to be 1e-3, which effectively uses 1000 steps. We also show how the number of steps (from 1000 to 20) would affect the quality in the new table 3.
>
> ### “In Appendix B.2, how does equation (34) equal to equation (35)? is this a typo?”
> We thank the reviewer for pointing out our typo (and your effort in checking the appendix is much appreciated!). We misplaced $s$ and $t$ in equation (34). We have fixed the typo in our revised version.
>
> ### Do we have the similar SDE formulations as in equation (5,6) for the discrete data since we already have the score function? Is it just a simple replacement of the proposed score function?
>
> This is an interesting question. However, we don’t have the SDE formulation for discrete data yet.
> - The score function we considered in discrete space is fundamentally different from the traditional score function defined in continuous space. In continuous space, the score function $\nabla \log \pi(x) = \frac{\nabla_i \pi(x)}{\pi(x)} \approx \frac{\pi(x + e_i) - \pi(x)}{\pi(x)} = \frac{\pi(x + e_i)}{\pi(x)} - 1$ characterizes the velocity of the movements, and can be either positive or negative. In discrete space, the “score function” in our context is the ratio $\frac{\pi(x + e_i)}{\pi(x)}$ characterizes the probability of jump, and is always positive.
> - In continuous space, the stochasticity is injected via adding gaussian noise. In discrete space, it is not clear how to inject additive noise to discrete variables. Our approach follows the standard Markov chain that injects the stochasticity via stochastic transitions between discrete states.
> Considering these two factors, the discrete diffusion model cannot be characterized in SDE formulation. Instead, we use the Markov jump process (also known as the continuous time Markov chain) as the diffusion model in discrete space.
>
> ### Code
> We have included the core code in the supplementary material. We will further refine the usage guidelines and upload pretrained models.

---

### Official Review · Reviewer_grv3 · 2022-10-24

**Confidence:** 3
**Correctness:** 3
**Technical Novelty And Significance:** 3
**Empirical Novelty And Significance:** 2
**Recommendation:** 6

**Clarity, Quality, Novelty And Reproducibility:**

Clarity

The submission is clearly written and easy to follow.

Novelty

The submission is novel by extending an existing framework.

Reproducibility

Implementation details are provided in the submission.

**Strength And Weaknesses:**

Strength

1. The submission is well written. The notations are well defined. The used concepts are background details are clearly provided in the submission, making the paper self-contained.

2. The empirical analysis is thorough and clear. The protocol, competing methods, expected results and the achieved results are clearly provided.

3. The methodology is novel. While the whole framework follows the existing framework of score-based generative models with SDEs, the proposed score function for discrete state variables is an extension to existing works.

Weakness

1. I am confused by the stochastic jump process. To the best of my knowledge, this terminology is more often used for jump processes like Poisson jump process. Specifically, it emphasizes more the discreteness of the path, instead of the discreteness of the variables: many stochastic jump processes have continuous-time variables. The model used in this submission is more a continuous-time discrete-state Markov process. Would it be possible for authors to clarify the rigorous definition of this terminology?

2. I am wondering whether we can "naively" use a score-based generative models with SDEs for discrete variables, by simply doing some discretization when conducting sampling? This might be a bad idea when the variable has only two categories, but what if it has more than 200 categories like the image example in the experiments? What is the problem of this simple solution? Answering this question can further motivate the submission.


**Summary Of The Paper:**

The submission considers score-based generative models with SDEs, but for discrete variables. The main challenge here is that the existing score-based methods do not apply to discrete variables since the log likelihood is not well defined. To deal with this issue, the authors use continuous-time discrete-state Markov process to model the forward process, with the according backward process also being a continuous-time discrete-state Markov process. Under this setting, a score-based estimation method is proposed for the diffusion model, with a sampling scheme for the reverse process.

The performance of the proposed method is thoroughly studied empirically, demonstrating an improved performance.

**Summary Of The Review:**

The submission is well written with lots of technical details. A new method is proposed with empirical results supporting the performance improvement. To the best of my knowledge, the submission extends the existing score-based method to discrete variables, which is a contribution to this area.

---

> ### Author Response · Authors · 2022-11-19
> **Response to Reviewer grv3**
>
> We thank the reviewer for providing helpful feedback! Please see our response below:
>
> ### “I am confused by the stochastic jump process…”
>
> We agree with the reviewer that the terminology “stochastic jump process” emphasizes the discrete movements. Hence, it can be used to describe both the discreteness of the path in ordinal space, such as a counting process, and the discreteness of the variable, such as a Markov jump process (also known as a continuous time Markov chain). As mentioned by the reviewer, the model we used in this submission is a continuous time Markov chain. The reason we used the terminology “stochastic jump process” is to make a comparison between “diffusion process” in continuous space. We admit that it can create ambiguity in some scenarios, so we change it with the terminology “continuous time Markov chain” in our revision.
>
> ### “...whether we can "naively" use a score-based generative models with SDEs…”
> In ordinary space, for example images in pixel representation, it is reasonable to use continuous relaxation (training continuous score-based model and rounding in sampling). But it is not straightforward to use continuous relaxation in non-ordinal discrete space.
> The motivation of this draft is to develop a discrete diffusion model for more general discrete spaces, such as, the vocabulary space for language or the code-book space for vector quantizations, where ordinal information is missing and we usually use a one-hot representation for variables. In these cases, a continuous relaxation could be hard to obtain, hence the traditional score-based models are not applicable. Therefore, we consider a continuous time Markov chain as the “diffusion” model,  and develop a corresponding categorical score matching method.

---

### Official Review · Reviewer_eZKv · 2022-10-25

**Confidence:** 4
**Correctness:** 4
**Technical Novelty And Significance:** 4
**Empirical Novelty And Significance:** 4
**Recommendation:** 10

**Clarity, Quality, Novelty And Reproducibility:**

I had no trouble understanding the paper and I think it is very nicely written. The approach is novel to me and it should be easy to reproduce the results.

**Strength And Weaknesses:**

Overall, I do not have much to say about this paper, besides that, I really really enjoyed reading it. It is written in an easy-to-follow way, even though continuous-time stochastic process modeling usually requires immense technical machinery. The approximate inference algorithm is derived in a principled way and the empirical results of the paper are very promising.

Some sugestions I have are:
- Including the result of equation (47) in the main text, would have made it easier for me to understand how the learning of the conditionals is related to the ratio in equation (10)
- In Sec 4.1. the transition matrix is given as an aside as $q_{t \mid s}(\cdot \mid \cdot)=\int_s^t \exp(Q_{\tau})\, \mathrm d \tau$. This is not correct as it is the solution to the linear ODE given in equation (7), for which the solution is given by the state-transition function, for which there is generally no decomposition as a matrix exponential, see, e.g., "Peano–Baker series", or, "ordered exponential".
- The results for reversing a CTMC are not new and well known in the context of continuous-time filtering and smoothing. The authors should consider citing the relevant work, see, e.g., [1-3], and the references therein.

[1] Anderson, Brian DO, and Ian B. Rhodes. "Smoothing algorithms for nonlinear finite-dimensional systems." Stochastics: An International Journal of Probability and Stochastic Processes 9.1-2 (1983): 139-165.
[2] Elliott, R. "Reverse-time Markov processes (Corresp.)." IEEE transactions on information theory 32.2 (1986): 290-292.
[3] Van Handel, Ramon. Filtering, stability, and robustness. Diss. California Institute of Technology, 2007.


**Summary Of The Paper:**

The paper presents a diffusion-type generative model for categorical data. For this, the authors derive a variational learning scheme to approximate the discrete state space analog of the score of the marginal likelihood. The authors present a forward noise and backward denoising model based on a continuous-time Markov chain model. For learning the authors present three different neural-network-based parametrizations for the variational approximation. The algorithm is extensively evaluated for synthetic and real-world data and shows very good results.

**Summary Of The Review:**

The paper presents a new algorithm for generative modeling with categorical data. It is very nicely written and it shows very competitive results.

---

> ### Author Response · Authors · 2022-11-19
> **Response to Reviewer eZKv**
>
> We thank the reviewer for the recognition of our work. We have carefully addressed the reviewer's suggestions in our revision. Please also see our response below:
>
> ### “Including the result of equation (47)...”
>
> We have added more explanation about how to use the conditionals to estimate the ratio in equation (12) in our revised version.
>
> ### “In Sec 4.1. the transition matrix…this is not correct…”
>
> We thank the reviewer for pointing out the neglect of noncommutative algebra in our expression. We have fixed our statement in our revised version.
>
> ### “The results for reversing a CTMC are not new…”
>
> We thank the reviewer for pointing out the missing references. We have cited the relevant work and revised our statement accordingly in our revised version.

---

### Official Review · Reviewer_4YQP · 2022-10-30

**Confidence:** 4
**Correctness:** 3
**Technical Novelty And Significance:** 2
**Empirical Novelty And Significance:** 2
**Recommendation:** 5

**Clarity, Quality, Novelty And Reproducibility:**

This could be solid paper if the weaknesses are sufficiently addressed. Clarity is generally good except Section 4 (see above comments). The work is original in applying discrete variants of score matching to diffusion models, although techniques mostly follow from prior work on ratio matching and the discrete diffusion model formulation by Campbell et al.

Minor:

* eq. (11), the first \approx should be identity, and the second identity should be \approx

**Strength And Weaknesses:**

## Strengths

* This paper is well-motivated. A well-known fact in the diffusion model community is that score-based objectives often lead to better likelihoods than the ELBO. It is natural to ask if this holds for discrete data. However, the score-based interpretation was overlooked in the prior literation of discrete diffusion models. They mostly mentioned that scores are undefined in this case but ignored the fact that Hyvärinen had written about discrete generalization of score matching (called ratio matching) in his own work.

* The proposed learning objectives are technically sound. As far as I checked, the derivations are correct. Modeling the complete conditionals also seems a very natural choice to me.

## Weaknesses

* The paper claims they extend score matching to general categorical data in several places. However, this is not true given the prior work on ratio matching and generalized score matching. The discrete analogy of score has also appeared in these works. From this perspective, the technical novelty seems to be quite limited.

* Although the motivation suggests that using score-based loss function should work better than ELBOs, the results on image data indicates the opposite. On simple datasets like CIFAR10, the proposed method was outperformed by prior discrete diffusion models, including D3PM, Campbell et al.

* One reason the paper uses for explaining the superiority of continuous diffusion models over discrete ones is that score matching for discrete variables does not benefit from prior knowledge about ordinal discrete data. However, I believe this is not the case for the "discrete score" definition in (11), which leverages the "sparse neighborhood" knowledge by only looking at states that differ by +-1.

* Section 4 is confusing. I do not get the reasoning "Such a jump rate depends on both the time t and ... Hence, it is inefficient to simulate the reverse time process X_t exactly".

**Summary Of The Paper:**

There is a recent emergence of interests in adapting diffusion models to handle discrete data and this work adds to that literature. To understand the main difference from prior work, it is important to note that there are two (almost) equivalent frameworks for learning diffusion models: the variational lower bound framework and the denoising score matching framework. Prior work on discrete diffusion models have mostly focused on the variational lower bound framework because score functions are not defined in discrete domains. This paper replaces the continuous score with a discrete analogy which were studied in previous work on ratio matching/generalized score matching. The authors then design a transformer-based architecture to model this "discrete score" (which is essentially all complete conditionals at time t). Experiments show that the proposed discrete score matching objective is effective in fitting discrete energy-based models and diffusion models, with comparable or worse results on image data but better results on music data.

**Summary Of The Review:**

The work investigates an under-explored direction for developing discrete diffusion models. However, the paper is not ready yet due to lack of necessary theoretical developments to address the major problems and insufficient empirical evidence for supporting their claims.

---

> ### Author Response · Authors · 2022-11-19
> **Response to Reviewer 4YQP**
>
> We thank the reviewer for providing helpful feedback! Please see our response below:
>
> ### "...Discrete analogy of score matching has appeared in previous works..."
>
> [1] defines the score matching (or ratio matching in binary discrete case) for *binary* case only, and cannot be straightforwardly extended beyond the case. For other generalized score functions in discrete space, for example [2] define the “score function” in ordinal discrete space for different purposes, there is no concept of score matching to learn the score function.
> In our work, we extend the score matching for categorical (i.e. beyond binary/ordinal random variables) random variables by minimizing the cross entropy of the conditional marginal distribution. The resulting categorical score matching in Prop 3.4, as an extension to the binary ratio matching of [1], is new and has never appeared in the literature, to the best of our knowledge.
>
> ### "...Experiment comparison on cifar10..."
> We want to kindly point out that the performance can be quite different depending on the space the algorithms operate on, and it is common practice to compare algorithms in the same state space. For example, D3PM achieves much worse performance in the categorical discrete space, compared to its performance in ordinal discrete space.
> We have conducted more experiments to make an *apples to apples comparison* in the categorical discrete space, where all the methods use the same transformer architecture and the same vector quantization, and operate in the code-book vocabulary space that is categorical without any ordering information. We included the results below for convenience:
>
> | Methods | IS $\uparrow$ | FID $\downarrow$ |
> |---|---|---|
> | D3PM* [4] | 8.85 | 16.47 |
> | $\tau$-LDR* [3] | 5.71 | 40.06 |
> | SDDM | **8.98** | **12.23** |
>
> One can see that the proposed SDDM consistently improves both the IS and FID scores under the same learning setting.
>
> > *We re-implemented both baselines to run in the same VQ space with the same architectures as the proposed SDDM. We tried our best to tune \tau-LDR with different learning rate schedules, different approximations (one-pass, two-passes as denoted in their paper).
>
> ### "...Discrete score in (11) can leverage the sparse neighborhood knowledge by only looking at states differ by +/- 1 ..."
> For generality consideration, our proposed framework never leverages the sparse neighborhood structure.
> The equation (11) in the original submission only provides a conceptual understanding of what a score function should look like in discrete space. In general discrete space, for example a finite graph, or a categorical space $\mathcal{X} = \{1, …, C\}^d$, the +/- 1 neighborhood might not exist. To elevate binary ratio matching to a general framework, we consider a categorical space, and accordingly the conditional distribution is defined on all possible $x \in \{1, …, C\}$. In this case, we have a full neighborhood and a state $c$ can jump to any other $c’ \in \{1, …, C\}$ in a single jump.
> Hence, it is easy to see that we do not leverage ordinal prior knowledge. This can also be verified in our experiments, with the fact that the neighborhood +/- 1 doesn’t even make any sense in the codebook space of vector quantization models.
> With that said, we admit that the equation (11) in the original submission can be confusing if one did not read the paper carefully. We have deleted it in our revised version, and directly give the definition of categorical ratio in the new equation (11).
>
> ### "...Section 4 is confusing..."
> In equation (18), we decompose each dimension of the Markov jump process $X_t$ into sub-processes $X^d_t$ to accelerate simulation of the process. However, in equation (18), one can notice that the rate $R^d_t(x, y; \theta) = \frac{p_t(X^d_t=y^d|x^{\backslash d}_t; \theta)}{p_t(X^d_t = x^d_t| x^{\backslash d}_t; \theta)} Q_t(y, x_t)$ for $X^d_t$ depends on the $x^{\backslash d}_t$, the value of the other dimensions. Once the value $x^{d’}$ in another dimension $d’$ changes, the rate $R^d$ for sub-process $X^d$ also changes. Hence, we cannot simulate each sub-process $X^d_t$ in parallel in exact sampling. That is why we say exact sampling is inefficient.
> We have revised the corresponding section to make this point more clear.
>
> **Reference**
> >[1] Aapo Hyvarinen. Some extensions of score matching. ¨ Computational statistics & data analysis, 51 (5):2499–2512, 2007\
> [2] Yang, J., Liu, Q., Rao, V. and Neville, J., 2018, July. Goodness-of-fit testing for discrete distributions via Stein discrepancy. In International Conference on Machine Learning (pp. 5561-5570). PMLR. \
> [3] A Continuous Time Framework for Discrete Denoising Models, Campbell et.al, 2022.\
> [4] Structured Denoising Diffusion Models in Discrete State-Spaces, Austin et.al, 2021.

---

### Author Response · Authors · 2022-12-06
**a kind reminder on reviewer-author discussions**

Dear Reviewers,

Thanks again for your effort and valuable feedback during the review process. We have made our response and revised the paper accordingly. Since the deadline for reviewer-author discussion is approaching, we would like to kindly remind you to please let us know if your concerns have been resolved, or if you still have any questions that we can address during this period.

Thank you so much for your understanding!

Best,

Authors

---

### Comment · Area_Chair_dQ57 · 2022-12-14
**Mixed reviews.**

Dear reviewers,
looks like the range of opinions on this paper varies greatly. Let's try to come to a conclusion on this paper.

Please have a look at the other reviews and comment on your peers.

4YQP, please have a look at the author response to see if your concerns have been addressed.

Cheers!

---

### Decision · Program_Chairs · 2023-01-20

**Decision:**

Accept: poster

**Justification For Why Not Higher Score:**

Overall the paper is a marginal accept. Two reviewers have increased their score during the rebuttal phase towards marginally above the acceptance threshold.

**Justification For Why Not Lower Score:**

Only one review has voted towards rejection. As we did not find evidence for the negative criticism, there is no reason to reject the paper.

**Metareview: Summary, Strengths And Weaknesses:**

The paper presents a generative diffusion model for categorical data. The paper claims to propose the first categorical score matching method and the first application of score matching to diffusion models.

Three reviewers vote for accepting the paper. The fourth reviewer criticises limited technical novelty and votes for reject.
pro:
- The reviewers find the main idea of using score-based objectives instead of ELBO surrogates well-motivated and novel.

The following weaknesses were identified by the reviewers:
-	One reviewer states that categorical score matching is not novel. (authors extend to categorical – not novel) but no reference is provided by the reviewer. The authors have refuted this criticism to the best of their knowledge. Unfortunately, the reviewer did not respond to the authors comments.
-	No experiment showing that continuous time formulation is better. This criticism could be addressed during the rebuttal phase, as the authors have provided additional experiments that show that the continuous time formulation indeed outperforms simpler discrete-time models.
- computationally demanding comparing to existing methods and prevents it from directly modeling the CIFAR10 data with resolution as small as 32x32. The inefficiency of the method was acknowledged by the authors.

After the rebuttal phase only the third one remains. Yet, the reviewers find that the methodical contribution outweighs the inefficiency of the method.


**Note From Pc:**

if the above contains the word "oral" or "spotlight" please see: "oral" presentation means -> notable-top-5% and "spotlight" means -> notable-top-25%. As stated in our emails, we are disassociating presentation type from AC recommendations

**Summary Of Ac-Reviewer Meeting:**

We went through the negative review. The most negative criticism was that categorical score matching is not novel. Yet, as we could not find a reference to prior work on categorical score matching, and as the authors also refute this claim to the best of their knowledge, we assume that the criticism is not valid. Unfortunately, the critical reviewer did not respond to the authors' comments.